# Phenotypic Differences and Physiological Responses of Salt Resistance of Walnut with Four Rootstock Types

**DOI:** 10.3390/plants11121557

**Published:** 2022-06-13

**Authors:** Xinying Ji, Jiali Tang, Wei Fan, Baoxin Li, Yongchao Bai, Junxing He, Dong Pei, Junpei Zhang

**Affiliations:** 1State Key Laboratory of Tree Genetics and Breeding, Key Laboratory of Tree Breeding and Cultivation of the State Forestry and Grassland Administration, Research Institute of Forestry, Chinese Academy of Forestry, Beijing 100091, China; jixinying111@163.com (X.J.); tangjl202205@163.com (J.T.); libaoxin94@163.com (B.L.); baiychao@163.com (Y.B.); hjx2648496184@163.com (J.H.); pei.dong@caf.ac.cn (D.P.); 2State Key Laboratory of Tree Genetics and Breeding, Chinese Academy of Forestry, Beijing 100091, China; fanwei-0551@163.com

**Keywords:** walnut rootstock, salt stress, growth characteristics, physiology and biochemistry, principal component analysis

## Abstract

Walnut is one of the world’s four largest nuts. Currently, the bottleneck in walnut breeding is the production of resistant variants. Soil salinization is a global problem, and the use of salt-tolerant rootstocks is a basic strategy to overcome the challenge of sustained walnut production. Providing a scientific basis for the selection of walnut salt-tolerant rootstocks is possible by studying the physiological and biochemical response characteristics and salt tolerance variations of different walnut genotypes under salt stress. In the present study, seedlings of four genotypes of walnut rootstocks, including J_1_ (*Juglans hindsii*), J_2_ (*J. mandshurica*), J_3_ (*J. regia* × *J. mandshurica*), and J_4_ (*J. regia* × *J. hindsii*), were employed as test materials to conduct a 28-day pot experiment under NaCl stress with five NaCl concentrations (0, 50, 100, 200, and 300 mmol/L). Under different NaCl treatment concentrations, seedling morphology, growth indices, chlorophyll content, photosynthetic parameters, relative electrical conductivity (REC), malondialdehyde (MDA), proline (Pro), soluble sugar (SS), and the activity of superoxide dismutase (SOD) and peroxidase (POD) in the leaves were examined. Salt stress altered the morphological characteristics and growth indices of seedlings from four genotypes to varying degrees. In addition, according to the analysis of physiological and biochemical data, salt stress had a considerable impact on both the physiological and biochemical processes of seedlings. Salt stress decreased the chlorophyll content and photosynthetic parameters of four genotypes, the REC, MDA content, Pro content, and SS content of each genotype increased by different degrees, and the enzymatic activities showed different trends. The salt tolerance of rootstocks was evaluated thoroughly using principal component analysis and membership function analysis based on the 16 parameters. The results of a comprehensive evaluation of salt tolerance showed that the order of salt tolerance of the four genotypes was J_4_ > J_1_ > J_3_ > J_2_, which corresponded to the order of the morphological symptoms of salt injury. In conclusion, J_4_ has strong salt tolerance and is an important germplasm resource for walnut salt-tolerant rootstock breeding.

## 1. Research Background

Soil salinization has become a global resource and environmental crisis, and it is one of the main factors restricting the development of agriculture and forestry [1]. According to statistics, the total area of all types of saline-alkali land in China is approximately 9.9 × 10^7^ hm^2^, accounting for nearly 10.3% of the total land area in China, and involves mainly the northeast, northwest, inland, and coastal regions of North China [2]. In China, saline-alkali soil is an important land reserve resource. Renovating and rationally utilizing saline-alkali soil is critical for agricultural productivity and sustainable development of the ecological environment [3].

Sodium chloride (NaCl) is one of the most common and widely distributed salts in the soil; Cl^−^ and Na^+^ in soil salt are lethal to plants and cause single salt toxicity [4,5]. Salt stress causes plant damage through a complex physiological and biochemical process. It is believed that salt stress causes damage to plants in three aspects: osmotic stress, ion toxicity, and nutritional imbalance [6]. Initially, plants are susceptible to osmotic stress, which inhibits water absorption by the roots, and then affects the transpiration rate and photosynthetic rate of plants, thereby, inhibiting plant growth and development. Under salt stress, the accumulation of Na^+^ and Cl^−^ in plants can affect the osmotic pressure and acid-base balance of plant cells, destroy cell membrane structure, and lead to metabolic disorders, and the competition between salt ions and nutrient elements leads to a nutritional imbalance in plants, inhibiting normal plant growth and development [7,8,9]. Scholars, both at home and abroad, have conducted numerous studies on the response of plants to salt stress and screened salt-tolerant resources by determining the external morphological characteristics of plants, the activities of antioxidant enzymes, and the changes in osmotic substances under salt stress [4,10,11,12,13,14,15,16]. An economical and effective method for screening salt-tolerant or salt-resistant plant materials is studying the physiological and biochemical indices of plants under salt stress [17,18].

Walnut (*Juglans regia* L.) is a perennial deciduous angiosperm that has gained increasing popularity in recent years owing to its various values [19,20]. It is an important nut and woody oil tree species, and the walnut industry has grown to be a pillar industry in many parts of China, playing an essential role in poverty alleviation projects. It has exceptional edible value and medicinal properties. It is also widely used as an excellent economical timber species with broad development prospects [20,21]. Grafting is a popular method of walnut breeding. Rootstock is the core source of resistance to various stresses required to sustain the growth and development of the whole plant [18]. The foundation of high-quality and high-yield walnut increased rootstock, and research has shown that walnut rootstock has an important impact on the growth and development, stress resistance, fruit yield, and quality of its grafted varieties [22,23,24].

Walnut stress resistance encompasses both biotic and abiotic stress resistance. Currently, research on walnut abiotic stress resistance has focused specifically on cold resistance [25,26] and drought resistance [20,27]. Plain areas are more suitable for walnut planting; however, to avoid occupying the agricultural land, expand the walnut planting area moderately, and obtain more economic benefits, many difficult sites with better conditions, such as saline-alkali land, have also begun to develop the walnut industry by changing the land to suit trees or changing trees to suit the land. Therefore, studying and screening rootstock varieties with strong salt tolerance is critical for adjusting the agricultural industrial structure, promoting agricultural efficiency, increasing farmers’ income, and rural revitalization. However, only a few studies have explored the salt tolerance mechanism of walnut rootstocks. Therefore, the present study explored the growth, physiological, and biochemical reactions of four walnut genotypes under salt stress to determine their tolerance to salt stress, clarify the physiological and biochemical mechanisms of salt tolerance, and screen walnut rootstock resources with strong salt tolerance, providing a scientific basis for the cultivation and application of walnut in salinized areas.

## 2. Materials and Methods

### 2.1. Plant Materials and Growth Conditions

The experiment was conducted in the greenhouse of the Chinese Academy of Forestry, China (Latitude 40°0′10″ N, longitude 116°14′38″ E, altitude 61 m). Four walnut seed genotypes of J_1_ (*J. hindsii*), J_2_ (*J. mandshurica*), J_3_ (*J. regia* × *J. mandshurica*), and J_4_ (*J. regia* × *J. hindsii*) were tested. J_1_ (*J. hindsii*), a tree species introduced to China, is native to the United States and is an excellent rootstock for grafted seedlings of walnut and black walnut. J_2_ (*J. mandshurica*) is often used as a rootstock for walnut grafting in northern China. J_3_ (*J. regia* × *J. mandshurica*), a natural hybrid of walnut and *J. mandshurica*, demonstrates a strong growth potential, strong resistance to adversity, and beautiful texture of nuts; it is not only used as the breeding material for cold and disease resistance of walnut plants in northern China but also in the production of Wenwan Walnut. J_4_ (*J. regia* × *J. hindsii*), a hybrid of walnut and *J. hindsii*, possesses the characteristics of barren resistance, strong stress resistance, fast growth rate, and high grafting affinity. On 2 January 2021, seeds with uniform size and no pests and diseases were selected and placed in 60-L buckets, to which clean water was added (the water level was kept higher than the level of seeds in the bucket). The seeds were soaked for 7 days, and the water was changed once a day. After the water was fully absorbed, the seeds were stored in the sand and stratified to accelerate the germination process. On 9 April 2021, the seeds that had been stored in the sand for 3 months were sown in plastic pots (18-cm diameter, 25-cm height, each containing one seed). The seeds were cultured in the greenhouse at an average temperature of 25 °C and not higher than 30 °C during the day and not lower than 14 °C at night, with the transmittance being 50–60% and the average relative humidity being 55–85%. The substrates used for culturing were peat soil, vermiculite, and perlite (3:1:1 *v*/*v*). On 22 April, the seeds began to emerge, and the seedlings were maintained and managed normally during the seedling growth period.

In late June, the seedlings were cultivated for approximately 2 months. The average plant height of each seedling was 45 cm, the average ground diameter was 6.4 mm, and the average number of compound leaves was 6. Healthy seedlings with consistent growth were selected for 28 days (1–28 July) under NaCl stress. A total of 5 salt gradients (0, 50, 100, 200, and 300 mmol/L) were set. Each gradient was divided into 3 groups of 6 plants each, and a total of 90 plants were selected for each genotype. The NaCl solution of each gradient was applied to all walnut genotypes during the morning (8:00–10:00 AM) on the same day. Plants were irrigated once a week with 300 mL applied per pot for a total of four times. To prevent leakage of the salt solution, a suitable-sized tray was placed at the bottom of the pot and the salt solution was returned from the tray to the pot. Routine management of the seedlings was performed during the experiment. After NaCl treatment, some seedlings were randomly selected from each group for biomass determination, and the 1–2 pairs of functional leaves at the middle and upper compound leaves of the other part of the seedlings were collected to measure the relevant physiological and biochemical indices.

### 2.2. Determination of Growth Indicators

Before salt stress treatment, 3 seedlings showing consistent growth were randomly selected from each group and their seedling height *H*_0_ (measured with a tape measure, cm) and ground diameter *D*_0_ (measured with a vernier caliper, mm) were recorded. After 28 days of salt stress, the seedling height *H*_1_ and ground diameter *D*_1_ were measured again, and the increment in seedling height *H*_∆_ = *H*_1_ − *H*_0_ and ground diameter *D*_∆_ = *D*_1_ − *D*_0_ were calculated. After the treatment, the seedlings were removed from each treatment group pot and rinsed with tap water; surface water was wiped with an absorbent paper, and plant fresh weight (PFW) was measured on an electronic balance. The plants were then divided into aboveground and underground portions and placed in cowhide bags and then into an oven at 105 °C for 30 min; the oven temperature was set to 75 °C for drying to a constant weight, removed, and weighed again to determine the dry weight, followed by the use of the below calculation formula: plant dry weight (PDW) = dry weight of shoot (DWS) + dry weight of root (DWR). Additionally, morphological changes in the leaves were recorded at the end of the salt stress period.

### 2.3. Determination of Physiological and Biochemical Indicators

#### 2.3.1. Determination of the Relative Electrical Conductivity

Relative electrical conductivity (REC) was measured using a DDS-11C conductivity meter and calculated using the method described by Ghalati et al. [28]. Briefly, approximately 0.1 g of fresh leaf sample was incubated in a 100 mL of water bath (40 °C, 30 min), and the electrical conductivity of the samples was measured (R1). After heating the samples in boiling water for 15 min, the electrical conductivity of the samples (R2) was measured again. The REC was used to represent the cell membrane permeability. The calculation formula used was as follows:REC (%) = R1/R2 × 100(1)

#### 2.3.2. Determination of the Chlorophyll Content

The pigments of leaves were extracted with 95% (*v*/*v*) ethanol and assessed according to the method of Zhu et al. [18]. The chlorophyll a (Chl a) and chlorophyll b (Chl b) contents in with the extracts were determined at the maximum absorption peaks of 665 nm and 649 nm, respectively. The contents of Chl a, Chl b, and TChl were calculated as follows:Chl a (mg/L) = 13.95A_665_ − 6.88A_649_(2)
Chl b (mg/L) = 24.96 A_649_ − 7.32 A_665_(3)
TChl (mg/L) = Chl a + Chl b(4)

#### 2.3.3. Determination of Photosynthetic Parameters

The first and second pairs of functional leaves of the middle and upper compound leaves were selected, and the photosynthetic indices such as net photosynthetic rate (*P_n_*), stomatal conductance (*G_s_*), transpiration rate (*T_r_*), and intercellular CO_2_ concentration (*C_i_*) were measured using the Li-6400 photosynthetic apparatus. The measurement time a was 09:00–11:00 AM on a sunny day, the CO_2_ concentration was 400 μmol·mol^−1^, and the measured light intensity was 1200 μmol·m^−2^·s^−1^. A standard blade chamber and open-air path were adopted, and the flow rate was set to 500 μmol·s^−1^.

#### 2.3.4. Determination of the Malondialdehyde Content

The malondialdehyde (MDA) content was determined according to the thiobarbituric acid (TBA)-based colorimetric method [29]. Briefly, fresh leaves were collected and extracted in 5 mL of 5% (*w*/*v*) trichloroacetic acid (TCA). The supernatant was collected by centrifugation at 10,000 rpm at 4 °C for 10 min, then add 2 mL TBA to the supernatant and incubated at 100 °C for 30 min. The tubes were incubated in an ice bath for 10 min to stop the reaction, and the absorbance of the supernatant was read at 450, 532, and 600 nm. The MDA content (μmol/g FW) was calculated as follows:[6.45 × (A_532_ − A_600_) − 0.56 × A_450_] × V_2_ × V/(m × V_1_ × 1000)(5)
where A_532_, A_600_, and A_450_ represent the absorbance of the supernatant at 532, 600, and 450 nm, respectively; V represents the volume of extraction (mL), V_1_ represents the volume of extracted liquid reacting with TBA, V_2_ represents the total volume of extract and the TBA reaction solution, and m represents the fresh weight of the samples (g).

#### 2.3.5. Determination of the Osmotic Adjustment Substances

##### Determination of the Proline Content

The content of proline (Pro) was measured using the ninhydrin reaction method [30]. Briefly, 0.1 g of fresh leaves were homogenized in 2.5 mL of 3% sulfosalicylic acid solution, and the homogenate was centrifuged at 10,000 rpm for 5 min. The extracted solution (2 mL) was treated with 2 mL of acid ninhydrin and 2 mL of glacial acetic acid and heated for 1 h at 100 °C. Then, 4 mL of methylbenzene was added to the solution to extract the mixture. The absorbance of the chromophore containing toluene was recorded at 520 nm.

##### Determination of the Soluble Sugar Content

The soluble sugar (SS) content was determined using the anthrone-sulfuric acid method as described by Liu et al. [6] and Kong et al. [31], with slight modifications. Frozen leaf material was extracted with 5 mL of distilled water at 100 °C for 30 min, after which the supernatant was collected. This step was repeated twice, and the supernatant was collected. Then, 0.5 mL of the extract was mixed with 1.5 mL of distilled water, 0.5 mL of anthrone reagent (1 g anthrone and 50 mL ethylacetate), and 5 mL of concentrated sulfuric acid; the mix was immediately placed in a boiling water bath for 1 min. After cooling, the SS content was analyzed through UV spectrophotometry at 630 nm.

#### 2.3.6. Determination of the Activity of Antioxidant Enzymes

The extraction of antioxidant enzymes was assayed by the method of Khalid et al. [32] with some modifications. Briefly, 0.3 g of each sample was taken and homogenized in 3 mL of sodium phosphate buffer (pH 7.8) with a chilled mortar and pestle, followed by centrifugation at 10,000 rpm at 4 °C for 10 min to obtain the supernatant as a crude enzyme extract. The supernatant was collected and used to determine the superoxide dismutase (SOD) and peroxidase (POD) activities.

SOD activity was determined according to Liu et al. [6] and Khalid et al. [32], with slight modifications. The activity of SOD was determined by monitoring the decrease in absorbance (560 nm) of the nitrobluetetrazolium (NBT) by the enzyme. The SOD activity was calculated as U/g FW (unit of enzyme activity per gram of fresh weight). The reaction mixture contained 50 mM phosphate buffer (pH 7.8), 13 mM methionine, 75 μM NBT, 10 μM ethylene diaminetetraacetic acid (EDTA)-Na_2_, 2 μM riboflavin, and distilled water (15:3:3:3:3:2.5 *v*/*v*). Approximately 3 mL of the reaction solution was taken in test tubes in duplicate, to one tube, 0.05 mL of the enzyme extract was added, and to the other tube, the enzyme extract was replaced with the same amount of phosphate buffer, serving as the blank control group. The tubes were placed under 4000 Lx fluorescent lamps for 30 min. At the end of the reaction, the lamps were turned off and the tubes were incubated in the dark for analyses.

POD activity was determined using the guaiacol method [15,33]. Briefly, 28 μL guaiacol was added to 50 mL PBS (0.2 M, pH 6.0), and the mixture was heated and stirred until complete dissolution. After cooling, 19 μL (30%) of H_2_O_2_ was added to the mixture. The mixture was then stored in a refrigerator for further use. Then, 3 mL of the solution was taken and mixed with 40 μL of the enzyme solution. PBS served as the blank, and the absorbance of the solutions was measured at 470 nm. One unit of POD activity was expressed as the change of absorbance per min and was calculated as U·g^−1^·min^−1^ FW.

### 2.4. Statistical Analysis

The original data were analyzed by Microsoft Excel 2010, and the data were imported into SPSS 23.0 software for ANOVA and Duncan multiple comparisons. Two-way analyses of variance (ANOVA) were performed to detect the effects of genotype and salinity, and their interactions. All statistical effects were considered significant at *p* < 0.05, and the index change map was analyzed and plotted by Origin 2018.

The measured indices were analyzed and screened by principal component analysis, and the data were comprehensively analyzed using the fuzzy mathematical membership function method to evaluate the salt tolerance of the four walnut genotypes.

The calculation method of membership function value was as follows: *R*(*X_i_*) = (*X_i_* − *X_min_*)/(*X_max_* − *X_min_*), and inverse membership function value: *R*(*X_i_*) = 1 − (*X_i_* − *X_min_*)/(*X_max_* − *X_min_*), where *X_i_* is the measured value of the index, and *X_min_* and *X_max_* are the minimum and maximum values of a certain index of all tested materials, respectively. Salt tolerance was determined by comparing the average values of the membership function values of all the measured indices of each genotype; the higher the average value, the stronger is the salt tolerance.

## 3. Results

### 3.1. Effects of Salt Stress on External Morphological Characteristics of Seedlings

The damage symptoms of four walnut genotypes appeared gradually with increasing salt concentration, however, significant differences were noted across genotypes (Figure 1, Table 1). At the NaCl concentration of 50 mmol/L, symptoms such as yellowing of the plants’ lower leaves and withered and curled leaf margins of the upper leaves emerged, and the yellowing, withering and curling of J_2_ was more severe. The symptoms of yellowing, withering, and curling of leaves of all genotypes were exacerbated after treatment with 100 mmol/L NaCl, with the symptoms of J_2_ being most severe. At the NaCl concentration of 200 mmol/L, the leaves of J_2_ and J_3_ were severely withered and curled, and deciduous leaves appeared; the lower leaves of J_1_ and J_4_ were yellowed, the middle and upper leaves were withered and curled, and a small number of leaves fallen off. When the NaCl treatment concentration reached 300 mmol/L, a large number of J_2_ leaves withered, curled, and fell off, whereas the number of withered and curled leaves in the middle and upper parts of J_4_ increased, and a small number of leaves fell off. During the period of NaCl stress, the salt stress symptoms of J_4_ appeared the latest compared with the other three genotypes. Therefore, the salt tolerance of the four genotypes can be preliminarily evaluated from the phenotypic symptoms as follows: J_4_ > J_1_ > J_3_ > J_2_.

### 3.2. Effect of Salt Stress on Plant Growth

As shown in Figure 2, the growth indices differed remarkably between the genotypes under salinity stress conditions. With the increase in NaCl concentration, the *H*_∆_, *D*_∆_, DWS, PFW, and PDW of the four genotypes exhibited a declining trend. At 50 mmol/L, the aforementioned growth indices of J_2_ significantly decreased compared with those of the control, but did not differ significantly from those of the control in J_4_. At -100 mmol/L, the aforementioned indices of $J_{3}$ were significantly different from those of the control, and the *D*_∆_, PFW, and PDW were significantly different from those of the control in J_1_ and J_4_. The growth indices fell to a larger extent as the NaCl concentration increased. When the NaCl concentration was 300 mmol/L, the *H*_∆_, *D*_∆_, DWS, PFW, and PDW of J_1_ decreased by 72.61%, 78.33%, 55.31%, 63.73%, and 63.26%, respectively, compared with those of the control. These growth indices of J_2_ decreased by 76.11%, 85.92%, 76.53%, 61.51%, and 74.99%, respectively. Compared with those of the control, the *H*_∆_, *D*_∆_ DWS, PFW, and PDW of J_3_ decreased by 75.61%, 82.08%, 76.18%, 72.39%, and 73.85%, respectively, and the growth indices of J_4_ decreased by 56.21%, 60.13%, 27.13%, 31.84%, and 34.65% respectively. Under high salt stress, the seedling growth of four walnut genotypes was seriously inhibited, among which J_2_ and J_3_ were seriously inhibited. Furthermore, the PFW and PDW were significantly affected by the interaction of genotype × salinity (Table 2).

### 3.3. Effect of Salt Stress on Chlorophyll Content

As shown in Figure 3, Chl a, Chl b, and TChl contents of all four genotypes declined with increasing salt concentration. The Chl a, Chl b, and TChl contents in J_1_, J_2_, and J_4_ did not differ significantly from those of the control at 50 mmol/L concentration; however, at the 100–300 mmol/L concentration range, the aforementioned indices of each treatment were different from those of the control. At 300 mmol/L, compared with the control, the contents of Chl a, Chl b, and TChl in J_1_ decreased by 58.12%, 45.18%, and 53.81%, respectively; the contents of Chl a, Chl b, and TChl in J_2_ decreased by 72.42%, 61.82%, and 68.43%, respectively; the contents of the corresponding indices of J_3_ decreased by 61.19%, 54.81%, and 59.30%, respectively; and the corresponding indices of J_4_ decreased by 55.40%, 47.23%, and 52.71%, respectively. The decrease in each index in J_2_ and J_3_ was significantly higher than that in the other two genotypes.

### 3.4. Effects of Salt Stress on Photosynthetic Gas Exchange Parameters

Photosynthesis is a metabolic process in plants that serve as the foundation for plant growth and development and a source of material and energy for plant growth. Figure 4 depicts the effects of different concentrations of salt stress on the photosynthetic parameters of the four walnut genotypes. The *P_n_*, *G_s_*, and *T_r_* of the four genotype seedlings showed a decreasing trend with increasing NaCl concentration, and NaCl treatment with different concentrations showed significant differences. When the NaCl concentration was 50 mmol/L, the aforementioned indices of the four genotypes decreased rapidly, with J_2_ exhibiting the highest reduction, and *P_n_*, *G_s_*, and *T_r_* decreased by 46.05%, 68.12%, and 58.95%, respectively, compared with the control. At 300 mmol/L, the aforementioned indices of the four genotypes decreased to the lowest; compared with the those of the control, the *P_n_*, *G_s_*, and *T_r_* of J_1_ decreased by 71.07%, 73.68%, and 78.88%, respectively; the *P_n_*, *G_s_*, and *T_r_* of J_2_ decreased by 91.70%, 84.06%, and 83.98%, respectively; the corresponding indices of J_3_ decreased by 85.57%, 84.96%, and 82.62%, respectively; and the corresponding indices of J_4_ decreased by 67.43%, 69.44%, and 72.95%, respectively. The decline in the aforementioned indices of J_2_ and J_3_ was higher than that in the other two genotypes. Under salt stress, when the NaCl mass concentration was 100–300 mmol/L, the *P_n_* of J_1_ and J_4_ changed slowly at a lower level compared with J_2_ and J_3_.

The *C_i_* of four walnut genotypes showed different trends with increasing salt concentration. With the increase in salt concentration, the *C_i_* of J_1_ and J_2_ increased, while that of J_3_ declined, and the *C_i_* of J_4_ first decreased and then increased. The results showed that the main limiting factors for the decreased *P_n_* caused by salt stress varied across four walnut genotypes. Furthermore, *P_n_*, *G_s_*, *T_r_*, and *C_i_* were significantly affected by the genotype × salinity interaction (Table 2).

### 3.5. Effects of Salt Stress on MDA Content and REC

At 200 mmol/L, the MDA content of J_1_, J_3_, and J_4_ was significantly higher than that of the control; whereas at 100 mmol/L, the MDA content of J_2_ was considerably higher than that of the control. At 200–300 mmol/L, the MDA content of J_2_ and J_3_ was higher and increased (Figure 5a). In addition, the MDA content was significantly affected by the interaction of genotype × salinity (Table 2). With an increase in the NaCl concentration, REC of different genotypes also increased, with the increase of J_2_ and J_3_ being the maximum. Under 300 mmol/L NaCl stress, compared with the control, the REC of the four genotypes increased by 1.73, 2.12, 1.87, and 1.71 times, respectively (Figure 5b). The results showed that salt stress damaged the cell membrane of four walnut genotypes to a certain extent, with J_1_ and J_4_ showing reduced damage compared with the other two genotypes.

### 3.6. Effects of Salt Stress on Osmotic Adjustment Substances

As shown in Figure 6a, the Pro content of J_1_ increased with increasing salt concentration, and significant differences in the Pro content were noted across treatments. The Pro content of J_2_ basically showed an upward trend; however, no significant difference was noted in the J_2_ Pro content across treatments. The Pro content of J_3_ also increased, and when the NaCl concentration reached 200 mmol/L, it differed considerably from that of the control. The Pro content of J_4_ first increased and subsequently declined, at the 0–100 mmol/L concentration range, the Pro content of J_4_ increased with the increase in salt concentration, whereas with further increase in the NaCl concentration, the Pro content decreased, and NaCl treatment with different concentrations showed significant differences.

The SS content of J_1_ and J_4_ first increased and subsequently decreased with increasing salt concentration. The SS content of J_1_ and J_4_ increased significantly at 50 mmol/L, and with further increase in the NaCl concentration, the SS content continued increasing; however, at 300 mmol/L concentration, the SS content decreased. Overall, the SS content of J_2_ and J_3_ exhibited an increasing trend. At the 0–200 mmol/L concentration range, the SS content of J_2_ and J_3_ increased slowly with the increase of salt concentration, whereas at 300 mmol/L, the SS content reached the maximum and was significantly higher than that of the control treatment; the SS content of J_2_ and J_3_ increased by 40.00% and 41.50%, respectively, compared with that of the control (Figure 6b). In addition, Pro and SS were significantly affected by the interaction of genotype × salinity (Table 2).

### 3.7. Effects of Salt Stress on Antioxidant Enzyme Activities

The SOD activity of four walnut genotypes basically first increased and then decreased under salt stress with different concentrations of salt; however, the extent of increase or decrease varied across different treatments and genotypes. The SOD activity in J_1_ and J_4_ increased progressively in response to increasing NaCl concentrations from 0 to 200 mmol/L, with plants exposed to 200 mmol/L showing the highest increase, and the SOD activity of J_1_ and J_4_ increased by 62.18% and 69.35%, respectively, compared with the control. J_2_ and J_3_ exhibited the maximum SOD activity at 50 mmol/L, which was significantly higher than the control. The SOD activity of J_2_ and J_3_ decreased again with continuous increasing salt concentration, and at 300 mmol/L, the SOD activity was lower than the control, although the difference was not significant. Under varying salt concentrations, the SOD activity of J_1_ and J_4_ was higher than that of the other two genotypes (Figure 7a). The four genotypes’ POD activity changed to different extents under different salt treatments. The POD activity of J_4_ increased with increasing salt concentration, peaking at 300 mmol/L, which was significantly higher than the control. J_1_, J_2_, and J_3_ increased initially and then declined with increasing salt concentration, reaching a maximum at 100, 50, and 200 mmol/L, respectively; However, at 300 mmol/L, the POD activity decreased to the control level, and the difference was not significant compared with the control. The fluctuation in POD activity of J_2_ was minimal among all treatments, and there was no significant difference between treatments, and the POD activity of J_2_ at each concentration was lower than that of the other three genotypes (Figure 7b). Furthermore, POD and SOD were significantly affected by the genotype × salinity interaction (Table 2).

### 3.8. Comprehensive Evaluation of Salt Tolerance

#### 3.8.1. Principal Component Analysis (PCA)

Through the principal component analysis of 16 indicators in four walnut genotypes grown in different salt treatments, four principal components with eigenvalues >1 were obtained, and the cumulative contribution rate reached 82.12%. On the PC1 (49.65%) and PC2 (18.57%) planes, four naturally segregating groups were formed. The first group is mainly characterized by photosynthetic parameters, namely net photosynthetic rate (*P_n_*), stomatal conductance (*G_s_*), transpiration rate (*T_r_*), and ground diameter increment (*D*_∆_). The second group is mainly characterized by growth traits, which include the dry weight of shoot (DWS), plant fresh weight (PFW), plant dry weight (PDW), seedling height increment (*H*_∆_), and total chlorophyll content (TChl). The third group is mainly characterized by osmotic adjustment substances and antioxidant enzyme activities, including proline (Pro), soluble sugar (SS), and the activity of superoxide dismutase (SOD) and peroxidase (POD). The fourth group mainly represents characteristics of membrane system traits, including relative electrical conductivity (REC), malondialdehyde (MDA), and intercellular CO_2_ concentration (*C_i_*) (Figure 8).

#### 3.8.2. Membership Function Analysis

Membership function analysis assessed 16 indices from four walnut genotypes, and the total average value was calculated to rank the salt tolerance (Table 3). The membership function value of J_4_ was the largest (0.668), followed by that of J_1_ (0.595), whereas the membership function value of J_2_ and J_3_ were 0.236 and 0.452, respectively. The results showed that the comprehensive order of salt tolerance of the four walnut genotypes was J_4_ > J_1_ > J_3_ > J_2_, which corresponded to the morphological performance of each genotype under salt stress and demonstrated the viability of the assessment approach.

## 4. Discussion

External morphological characteristics of plants can directly reflect the effects of salt stress on plants, and thus serve as an index of plant salt tolerance [10,34]. The changes in the plant growth indices and biomass are the comprehensive embodiment of plants to salt stress, and it is also the most intuitive index to judge the salt tolerance of plants [35]. By observing the external morphology of four genotypes after salt stress, it was found that the onset of stress symptoms occurred later in J_4_ than in the other three genotypes; after 28 days of stress, all genotypes showed symptoms of salt injury to varying degrees under different salt concentrations; however, the salt injury symptom of J_2_ was the most serious under all concentrations and that of J_4_ was less serious than that of other genotypes. Salt stress also inhibited the growth of four walnut genotypes, and *H*_∆_, *D*_∆_, DWS, PFW, and PDW of all genotypes decreased with increasing salt concentration. Low-concentration (50 mmol/L) salt stress had no significant effect on the growth of J_1_ and J_4_, but it did have an inhibitory role, with the effect on J_2_ growth being more significant. The four walnut genotypes’ growth was significantly inhibited by 300 mmol/L salt stress, with J_4_ having the smallest decline in growth indices under high salt stress. According to the phenotypic symptoms of each genotype, the order of the four walnut genotypes according to their salt tolerance was: J_4_ > J_1_ > J_3_ > J_2_.

Salt stress might inhibit and influence plant photosynthesis to some extent. First, chlorophyll is the most important pigment involved in photosynthesis in plants, and it is essential in organic compounds and light energy conversion [36]. Chlorophyll content can reflect the strength of photosynthesis under salt stress, which is one of the important physiological indices used to measure plant stress resistance, and its content has a direct effect on plants’ growth and development [30]. The contents of Chl a, Chl b, and TChl of four walnut genotypes decreased with the increasing salt concentration in this study. At the NaCl concentration of 50 mmol/L, Chl a, Chl b and TChl content of J_1_ and J_4_ exhibited no significant differences compared to the control group, indicating that low salt stress had a slight effect on the photosynthetic pigments of the aforementioned two walnut genotypes. Under high salt (300 mmol/L) stress, the photosynthetic pigment of four walnut genotypes was significantly lower than that of the control, with the above-mentioned indicators of J_2_ having a relatively large reduction. In general, a decrease of these pigments under salt stress is considered to be a result of slow synthesis or fast breakdown of the pigments in the cells [30]. Salt stress enhanced the synthesis of ROS, which caused lipid peroxidation and resulted in a large accumulation of MDA content. Hight concentration of MDA can promote the destruction of chloroplast membrane structure and accelerate the degradation of chlorophyll [29,37]. Chlorophyll synthesis was disrupted under NaCl stress, and the chlorophyll content decreased, which decreased the photosynthetic rate of plants, affecting plant growth and development and biomass accumulation. In J_4_ plants, the chlorophyll content decreased slightly under NaCl stress, the chlorophyll system loss was the least, and the salt tolerance was the best. Second, salt stress can affect plant photosynthetic gas exchange parameters. *P_n_* can directly reflect plant assimilation capacity per unit leaf area and is an important index for measuring plant photosynthetic capacity. The results of this experiment showed that with increasing salt concentration, the *P_n_*, *G_s_*, and *T_r_* of the four walnut genotypes showed a downward trend, which was significantly inhibited, and this phenomenon is consistent with the previous research results of 4 olive cultivars [38] and *Prunus* rootstocks [5] under salt stress. The decrease in *P_n_* of J_4_ under high salt stress was less than that in the other three genotypes, indicating that J_4_ has a stronger ability to produce organic matter under high salt stress. According to studies, the decrease of *P_n_* in plant leaves can be attributed to two categories: if the decrease in *P_n_* is accompanied by the increase of *C_i_*, the main limiting factor for photosynthesis is the nonstomatal factor; if both *C_i_* and *G_s_* decrease at the same time, the main limiting factor is stomatal [39,40]. In this experiment, with increasing salt concentration, the *P_n_* of J_1_, and J_2_ decreased gradually, accompanied by an increase in *C_i_*, indicating that the nonstomatal restriction was the primary cause of the decrease in *P_n_* in the aforementioned two genotypes. It may be due to the temporary enhancement of cell respiration under high salt stress, as well as the accumulation of excess salt ions in the cells, which destroyed the chloroplast structure and leaf photosynthetic organs, resulting in a decrease in the chlorophyll content and photosynthetic activity of leaves. In J_3_, the gradual decrease in *P_n_* was accompanied by a decrease in *C_i_*, indicating that stomatal limitation was the main factor for the decrease in *P_n_*. It may be inferred that osmotic stress caused by salt stress reduced *G_s_* and increased the resistance of ${\mathrm{CO}}_{2}$ diffusion from the outside to the cell, resulting in photosynthetic inhibition. During the stress period, the *C_i_* of J_4_ first decreased and subsequently increased, therefore, changing the photosynthesis limitation factor of J_4_ from stomatal to nonstomatal.

Studies have shown that under stress conditions such as salt stress, the membrane system of plants is destroyed, the selective permeability of cell membranes increases, and a large number of intracellular substances extravasate. The increased membrane permeability and electrolyte extravasation lead to increased REC in plants. Therefore, REC represents the degree of damage to the plant cell membrane under stress conditions [18,28]. MDA is a membrane lipid peroxidation and serves as a significant physiological marker indicating the degree of cell membrane damage and the strength of its response to adversity [41]. The contents of REC and MDA increased to varying degrees in this experiment, demonstrating that different concentrations of salt stress can cause lipid peroxidation of plant cell membranes and damage plant cell structures to varying degrees. J_1_ and J_4_ showed a lower degree of cell membrane lipid peroxidation and smaller increases in the REC and MDA contents. This finding indicated that under NaCl stress, J_1_ and J_4_ could better maintain the stability of the cell membrane structure and alleviate the oxidative damage to cells caused by high salt stress than J_2_ and J_3_.

Excessive salt ions in the soil limit soil water potential, making it difficult for plants to absorb water or water outflow, resulting in osmotic stress. Plants can regulate intracellular osmotic potential by synthesizing and accumulating organic osmolytes in excess under salt stress conditions to avoid osmotic stress damage, maintain osmotic balance, and ensure the normal physiological functioning of cells [42]. Pro as an osmotic adjustment substance in plants, can be used as a reference index for the degree of salt and alkali stress in plants. It can regulate osmotic potential, scavenge free radicals, and stabilize subcellular structures [29]. Some studies have shown a positive correlation between Proline accumulation and NaCl concentration in plants [4,30,43]. The contents of Pro in the four genotypes increased at different levels of salt stress in this experiment. The Pro content of J_1_ and J_4_ increased significantly under high salt concentration and the relative Pro content of these two genotypes was higher than that of the other two genotypes. This result showed that under high salt stress, J_1_ and J_4_ could deal with the damage induced by salt stress by accumulating more Pro, which enhances their adaptability to the NaCl stress environment. SS is an important osmotic regulatory substance in plants, regulating the change in osmotic potential in cells, and a carbon framework for intracellular macromolecules and an energy source in plants [44]. The SS content of four walnut genotypes increased in varying degrees with the increase in NaCl concentration in this experiment, which is consistent with the results of previous studies on the effect of salt stress on bean seedlings [45], *Oriza sativa* L. [46], and 4 kiwifruit genotypes [10]. Additionally, the SS content of J_1_ and J_4_ decreased under high salt stress, which might be because under high concentration salt stress, these two genotypes needed to consume more energy to maintain their resistance physiology.

When plants are stressed, the body produces reactive oxygen species (ROS) and other harmful substances, which cause plant senescence, oxidative stress, and other damages. When plants are subjected to salt stress, a substantial quantity of ROS accumulates, causing oxidative damage to plant cells. To limit the damage caused by ROS, plants produce a sophisticated and complex antioxidant system to eliminate excessive ROS, thus improving plant salt tolerance [44]. SOD and POD enzymes are components of the cellular defense system that maintain a metabolic balance [47]. SOD is an essential enzyme in plant antioxidant systems for scavenging oxygen free radicals, and it can decompose superoxide anion into singlet oxygen and H_2_O_2_ and resist ROS damage to the cell membrane. H_2_O_2_ retains strong oxidability, and POD enzyme is required for further decomposition into O_2_ and H_2_O [44,48]. The SOD enzyme activity of four walnut genotypes first increased and then decreased with increasing salt concentration, and the SOD enzyme activity of J_1_ and J_4_ was higher than that of the other two genotypes, which is consistent with the previous studies on 4 citrus rootstocks [15] and *Phoenix dactylifera* L. [49]. The SOD enzyme activities of four genotypes decreased under high salt stress, but the enzyme activities of J_1_ and J_4_ were higher than those of the control, suggesting that J_1_ and J_4_ could remove the damage caused by excessive ROS and alleviate the damage to cell membrane lipid peroxidation by maintaining a high SOD enzyme activity. The variation in the POD enzyme activity may reflect plant stress resistance [34,50]. In general, the higher the activity of the enzyme POD, the stronger the stress resistance of plants, and with the aggravation of stress, the POD activity steadily diminishes until inactivated [51]. The changes in the POD enzyme activity of four walnut genotypes showed different changing trends with increasing salt concentration. The POD activity of J_4_ increased with an increase in the salt concentration, and the differences across treatments were significant, with J_4_ demonstrating a strong salt tolerance. Some studies have shown that the higher the salt concentration, the greater the inhibition of plant growth and the higher the activity of antioxidant enzymes; however, when the salt concentration exceeds the tolerance range of plants, the antioxidant capacity of plants gradually diminishes [32,52,53]. The POD activity of J_1_ and J_3_ first increased and subsequently decreased; it increased significantly at 50 and 100 mmol/L concentrations, respectively, and remained at a high level. However, when the NaCl concentration increased to 300 mmol/L, the cells’ self-regulation ability was seriously limited, the cell membrane system was excessively damaged, and the POD activity of the two genotypes decreased to the control level. The POD enzyme activity of J_2_ also showed a trend of first increasing and then decreasing, at 50 mmol/L, the POD activity increased slightly, and there was no significant difference between the treatments and the control; and the POD activity of J_2_ under each salt concentration treatment was generally lower than that of the other three genotypes. Among the four genotypes, J_4_ exhibited higher antioxidant enzymes (SOD and POD) activities and showed strong salt tolerance.

At present, plant salt tolerance physiological research is mainly based on neutral salt NaCl stress, this study is also based on NaCl stress to explore the physiological and biochemical changes of four walnut genotypes. However, the large areas of saline-alkali land in China are complex saline-alkali land; and two major problems are soil alkalization caused by alkaline salts such as Na_2_CO_3_ and NaHCO_3_ and soil salinization caused by neutral salts such as NaCl and Na_2_SO_4_; and in actual saline-alkali land, the problems of soil salinization and alkalization often coexist. Furthermore, because plant salt tolerance is regulated by various trait genes, gene expression varies significantly under different ecological environment conditions and growth stages, and the salt tolerance performance may change. Thus, the salt tolerance of different walnut genotypes should be further studied in terms of physiological and ecological responses and gene network regulation mechanisms in the soil salt environment.

## 5. Conclusions

In the present study, we investigated the growth, physiological and biochemical responses of four different walnut genotypes against salt stress. According to the results, salt stress is reflected in obvious impairment in photosynthesis and inhibition of stomatal conductance, transpiration, and chlorophyll synthesis, with an increase in the concentration of MDA and REC highlighting the lipid peroxidation and oxidative damage of cell membranes. However, the adverse effects of salt stress could be mitigated via various internal mechanisms in the cell, such as the accumulation of antioxidant enzymes (SOD and POD) and osmolytes (Pro and SS) (Figure 9). The salt stress-induced diverse changes among all four walnut genotypes. Considering the results of principal component analysis and membership function, the comprehensive order of the four genotypes according to their salt tolerance was: J_4_ > J_1_ > J_3_ > J_2_. It was consistent with each genotype’s morphological performance under salt stress and demonstrated variances in growth characteristics, and physiological and biochemical responses of four walnut genotypes under different salt stress.

## Figures and Tables

**Figure 1 plants-11-01557-f001:**
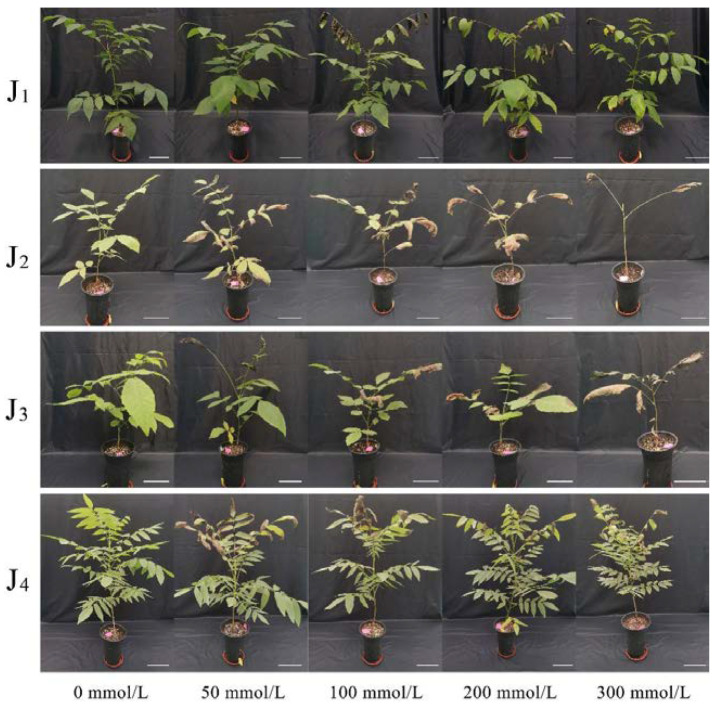
Comparison of the morphological performance of walnut rootstocks under saline irrigation treatment. Four walnut genotypes were grown under different NaCl concentrations (0, 50, 100, 200, and 300 mmol/L) for 28 days. Salt tolerance capacity was estimated after 28 days. Scale bar: 15 cm.

**Figure 2 plants-11-01557-f002:**
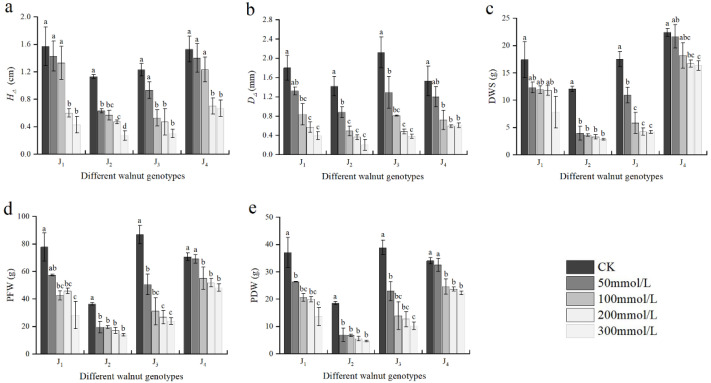
Effect of salt stress on the growth of walnut. (**a**) Seedling height increment (*H*_∆_), (**b**) ground diameter increment (*D*_∆_), (**c**) dry weight of shoot (DWS), (**d**) plant fresh weight (PFW), and (**e**) plant dry weight (PDW). Four walnut genotypes were grown under different NaCl concentrations (0, 50, 100, 200, and 300 mmol/L) for 28 days. Each value indicates the mean ± standard error (SE) of three biological replicates. Bars of each parameter labeled with different letters are significantly different (*p* < 0.05) by Duncan’s test.

**Figure 3 plants-11-01557-f003:**
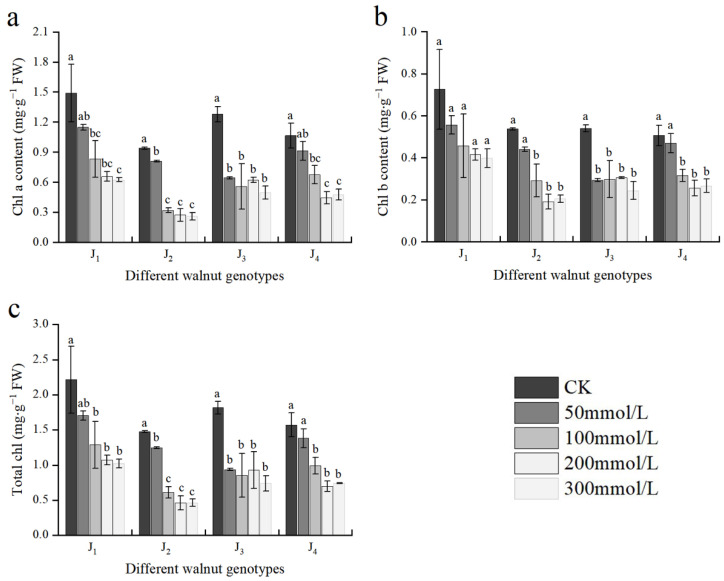
Effect of salt stress on the chlorophyll content of walnut. (**a**) Chlorophyll a content (Chl a), (**b**) chlorophyll b content (Chl b), and (**c**) total chlorophyll content (TChl) in the leaves. Four walnut genotypes were grown under different NaCl concentrations (0, 50, 100, 200, and 300 mmol/L) for 28 days. Each value indicates the mean ± standard error (SE) of three biological replicates. Bars of each parameter labeled with different letters are significantly different (*p* < 0.05) by Duncan’s test.

**Figure 4 plants-11-01557-f004:**
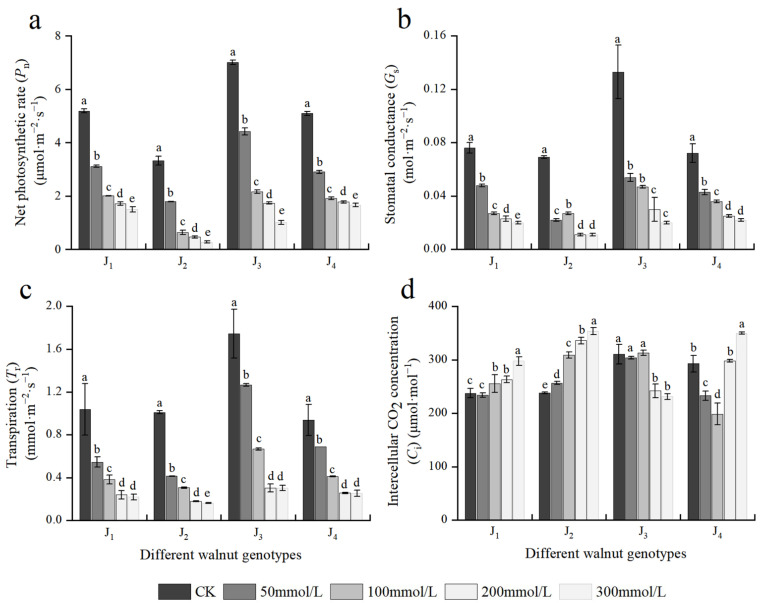
Effect of salt stress on the photosynthetic parameters of walnut. (**a**) Net photosynthetic rate (*P_n_*), (**b**) stomatal conductance (*G_s_*), (**c**) transpiration (*T_r_*), and (**d**) intercellular CO_2_ concentration (*C_i_*) in the leaves. Four walnut genotypes were grown under different NaCl concentrations (0, 50, 100, 200, and 300 mmol/L) for 28 days. Each value indicates the mean ± standard error (SE) of three biological replicates. Bars of each parameter labeled with different letters are significantly different (*p* < 0.05) by Duncan’s test.

**Figure 5 plants-11-01557-f005:**
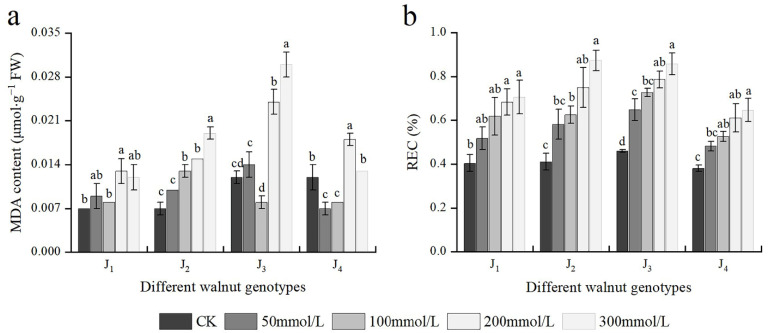
Effect of salt stress on (**a**) the malondialdehyde (MDA) content, (**b**) relative electrical conductivity (REC) of walnut. Four walnut genotypes were grown under different NaCl concentrations (0, 50, 100, 200, and 300 mmol/L) for 28 d. Each value indicates the mean ± standard error (SE) of three biological replicates. Bars of each parameter labeled with different letters are significantly different (*p* < 0.05) by Duncan’s test.

**Figure 6 plants-11-01557-f006:**
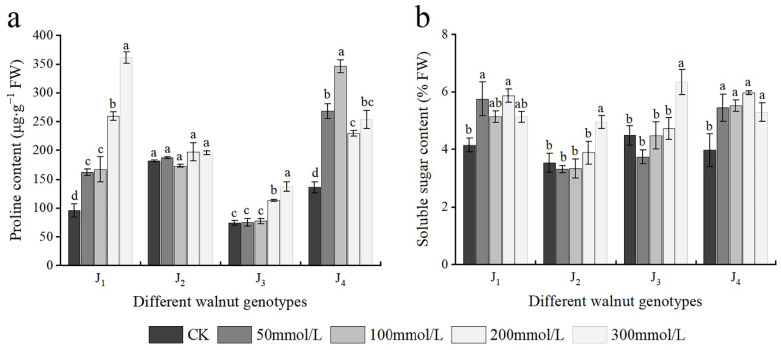
Effect of salt stress on the osmotic adjustment substances of walnut. (**a**) Proline (Pro) content, (**b**) soluble sugar (SS) content in the leaves. Four walnut genotypes were grown under different NaCl concentrations (0, 50, 100, 200, and 300 mmol/L) for 28 days. Each value indicates the mean ± standard error (SE) of three biological replicates. Bars of each parameter labeled by different letters are significantly different (*p* < 0.05) by Duncan’s test.

**Figure 7 plants-11-01557-f007:**
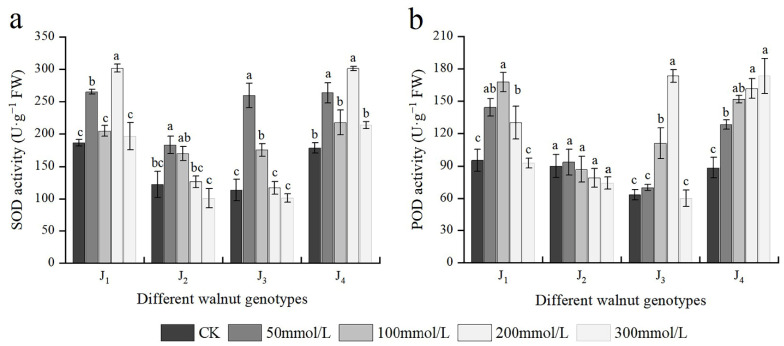
Effect of salt stress on the activities of antioxidant enzymes of walnut. (**a**) Superoxide dismutase (SOD) activity, and (**b**) peroxidase (POD) activity of the leaves. Four walnut genotypes were grown under different NaCl concentrations (0, 50, 100, 200, and 300 mmol/L) for 28 days. Each value indicates the mean ± standard error (SE) of three biological replicates. Bars of each parameter labeled with different letters are significantly different (*p* < 0.05) by Duncan’s test.

**Figure 8 plants-11-01557-f008:**
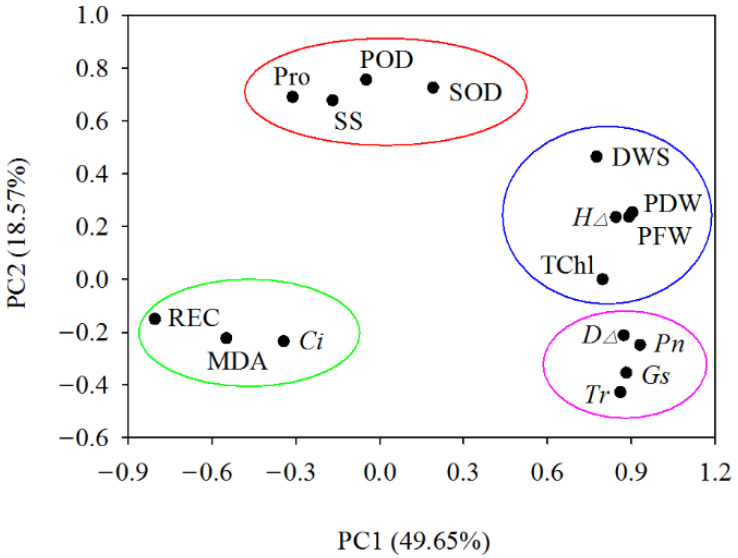
Principal component analysis of walnut genotypes under different salt concentrations. Drawing a scatterplot with PC1 as the X-axis and PC2 as the Y-axis, four naturally segregating groups were formed. Abbreviations of the corresponding indicators are shown in Figure 2, Figure 3, Figure 4, Figure 5, Figure 6 and Figure 7.

**Figure 9 plants-11-01557-f009:**
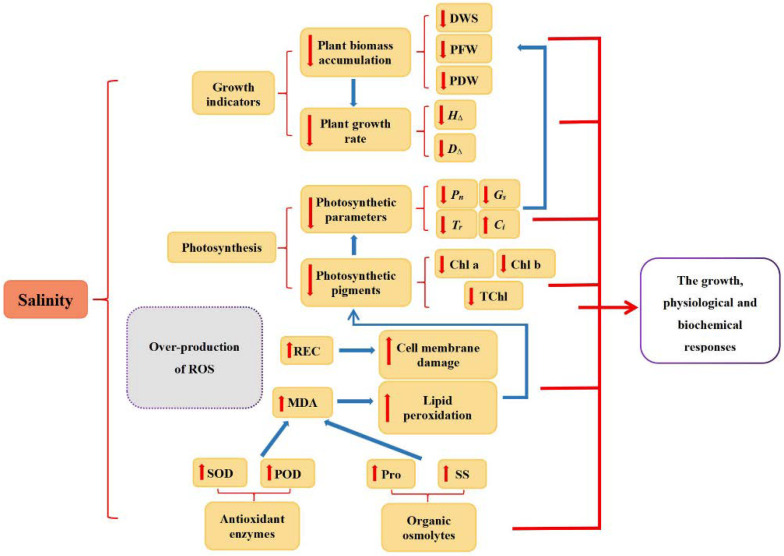
Schematic representation of salinity-induced growth inhibition. *H*_∆_,seedling height increment; *D*_∆_,ground diameter increment; DWS, dry weight of shoot; PFW, plant fresh weight; PDW, plant dry weight; Chl a, chlorophyll a content; Chl b, chlorophyll b content; TChl, total chlorophyll content; *P_n_*, net photosynthetic rate; *G_s_*, stomatal conductance; *T_r_*, transpiration; *C_i_*, intercellular CO_2_ concentration; MDA, malondialdehyde; REC, relative electrical conductivity; Pro, proline; SS, soluble sugar; SOD, superoxide dismutase; POD, peroxidase.

**Table 1 plants-11-01557-t001:** Morphological characteristics of 4 different walnut genotypes under NaCl stress.

NaCl Concentration /(mmol/L)	Morphological Characteristic of Plants			
	J_1_	J_2_	J_3_	J_4_
0 (CK)	Normal color of leaves and growth of plant	Normal color of leaves and growth of plant	Normal color of leaves and growth of plant	Normal color of leaves and growth of plant
50	Few leaves became yellow and withered	Partial leaves became yellow, withered and curled	Few leaves became yellow, withered and curled	Partial leaf margin withered and curled at upper of plant
100	Partial leaves became yellow, withered and curled	More leaves obvious became yellow, withered and curled	More leaves became yellow, withered and curled	Partial leaves withered and curled at upper of plant
200	More leaves became yellow, withered and curled, few leaves fallen off	Most leaves withered and curled, and leaves obvious fallen off	Leaves withered and curled seriously, partial leaves fallen off	More leaves withered and curled, few leaves fallen off
300	More leaves obvious became yellow, withered and curled, partial leaves fallen off	Most leaves withered, curled and fallen off	More leaves obvious withered, curled and fallen off	More leaves obvious withered and curled, partial leaves fallen off

**Table 2 plants-11-01557-t002:** Main and interactive effects of genotype and salinity on growth, and physiological and biochemical parameters of walnut.

Parameters	Main Effects		Interaction
	*Genotype(G)*	*Salinity(S)*	*G* × *S*
*H* _∆_	**0.000**	**0.000**	0.350
*D* _∆_	**0.013**	**0.000**	0.797
DWS	**0.000**	**0.000**	0.127
PFW	**0.000**	**0.000**	**0.005**
PDW	**0.000**	**0.000**	**0.034**
Chl a	**0.000**	**0.000**	0.389
Chl b	**0.000**	**0.000**	0.914
TChl	**0.000**	**0.000**	0.632
*P_n_*	**0.000**	**0.000**	**0.000**
*G_s_*	**0.000**	**0.000**	**0.000**
*T_r_*	**0.000**	**0.000**	**0.000**
*C_i_*	**0.000**	**0.000**	**0.000**
MDA	**0.000**	**0.000**	**0.000**
REC	**0.000**	**0.000**	0.900
Pro	**0.000**	**0.000**	**0.000**
SS	**0.000**	**0.000**	**0.001**
SOD	**0.000**	**0.000**	**0.000**
POD	**0.000**	**0.000**	**0.000**

Abbreviations of the corresponding indicators are shown in Figure 2, Figure 3, Figure 4, Figure 5, Figure 6 and Figure 7. Two-way ANOVAs were applied to evaluate the effects of different factors and their interactions. Significant values (*p* < 0.05) are shown in bold.

**Table 3 plants-11-01557-t003:** The average membership function values of the four walnut genotypes under salt stress.

Index	$J_{1}$	$J_{2}$	$J_{3}$	$J_{4}$
*H* _∆_	0.594	0.182	0.259	0.660
*D* _∆_	0.558	0.205	0.541	0.555
DWS	0.484	0.064	0.260	0.461
PFW	0.614	0.060	0.883	0.840
PDW	0.665	0.056	0.488	0.840
TChl	0.755	0.276	0.294	0.844
*P_n_*	0.719	0.037	0.861	0.715
*G_s_*	0.447	0.063	0.830	0.534
*T_r_*	0.277	0.067	0.888	0.370
*C_i_*	0.301	0.626	0.546	0.483
REC	0.571	0.392	0.198	0.760
MDA	0.835	0.592	0.361	0.699
Pro	0.581	0.545	0.040	0.743
SS	0.651	0.252	0.540	0.662
POD	0.663	0.276	0.299	0.730
SOD	0.812	0.192	0.294	0.844
Average	0.595	0.236	0.452	0.668
Rank	2	4	3	1

## Data Availability

Not applicable.

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
