# Peer review of "Phenotypic Differences and Physiological Responses of Salt Resistance of Walnut with Four Rootstock Types"

_plants, 2022, doi:10.3390/plants11121557_

Round 1

Reviewer 1 Report

The manuscript “Phenotypic Differences and Physiological Responses of Salt Resistance of Walnut with Four Rootstock Types” seems to be an excellent topic and time-worthy work indeed.In this study, the authors tried to explain some morphological and biochemical features of 4 walnut genotypes under salt stress.  However, some queries need to address to improve the manuscript. Some typo errors found in the whole manuscript, I strongly suggest revising and check the entire manuscript carefully, for which I recommend minor revision.

Line 333-334: Please rephrase the sentence.

Line 350-353: The information is not clear; please rephrase the sentence.

Line 456-460: One of the main reasons behind the decreased chlorophyll and net photosynthesis is the higher accumulation of MDA due to osmotic stress. https://doi.org/10.1371/journal.pone.0262099

It would be nice to make a schematic representation based on the results of the experiment. Please see the example…. https://doi.org/10.3390/plants10071313

Conclusion: It is better to avoid any general discussion in the conclusion part; please try to focus on the principal findings.

Reviewer 2 Report

The manuscript is well written and coherently presented, there are no methodological errors. 

Minor spell ckeck is needed all over the manuscript.

Several examples:

Line 45: NaCl ... widely distributed salts. Please correct it to ... widely distributed salt. 

Lines 73-75: resistance is too many times repeated. Please simplify where possibile. 

Lines 108-109: please rewrite the description of the greenhouse conditions (temperature not higher and not lower than...)

Line 118: please change was with were

Line 119: please rewrite the NaCl solution treatment. The plants were treated 4 times with the used solutions (5 gradients), or the final concetration was the mentioned ones?

Line 153: content in the extracts. We propose to change in with of the extracts

Line 161-162: please change the full stop signs.

Questions and recommendations regarding the figures and tables used in the manuscript

Table 1: do you consider that it is possibile to simplify (leaf colouring, leaf margin colouring, curling, leaf falling)/even quantify (+,++, +++) this table to be more traceable. 

Figure 2, 3, 5, 7 are different from figures 4 and 6.  Why did you opted for different chart types? The treatment was similar, the plant material similar, therefore the trends would be more visible for all studied parameters/variables on the same type of chart. 

Reviewer 3 Report

The manuscript is interesting and well written. I have only several comments and suggestions.

In the material and methods section, the sentence on drying of samples should be reformulated. As it is currently written it can be understood that all the sampled material was dried in the oven, but the biochemical analyses were performed on fresh material. It would be necessary to explain whether from each seedling only a fraction was dried, or other seedlings were used for biochemical analysis.

I suggest representing all data in the form of bar graphs (like the one in Figure 6) using different colours. This type of graph is more informative as it includes the letters of the post hoc tests. An alternative is to present the data with statistics in a table.

I suggest including only the graph with the first and second principal component, and instead of Figure 8B, to include the scatter plot of the PCA scores of the cultivars. 

A table including the significance of the two factors (Treatment and Genotype) and their interaction would improve the interpretation of the results.

Author Response

请参阅附件。
